# The Impact of Angiogenesis in the Most Common Salivary Gland Malignant Tumors

**DOI:** 10.3390/ijms21249335

**Published:** 2020-12-08

**Authors:** Despoina Pouloudi, Aristoteles Sotiriadis, Margarita Theodorakidou, Panagiotis Sarantis, Alexandros Pergaris, Michalis V. Karamouzis, Stamatios Theocharis

**Affiliations:** 1First Department of Pathology, Medical School, National and Kapodistrian University of Athens, 11527 Athens, Greece; d.v.pouloudi@gmail.com (D.P.); ari.sotiriadis@gmail.com (A.S.); mtheodorakidou@gmail.com (M.T.); alexperg@yahoo.com (A.P.); 2Molecular Oncology Unit, Department of Biological Chemistry, Medical School, National and Kapodistrian University of Athens, 11527 Athens, Greece; psarantis@med.uoa.gr (P.S.); mkaramouz@med.uoa.gr (M.V.K.)

**Keywords:** salivary gland tumor, angiogenesis, angiogenic factors, anti-angiogenic factors, histotypes, therapy

## Abstract

Salivary gland carcinomas (SGCs) represent a group of rare tumors, with complete surgical resection being the main treatment option. Therapeutic armory for cases of locally aggressive, recurrent, and/or metastatic SGCs, though, remains poor since they exhibit high rates of resistance to systematic therapy. Angiogenesis is considered one of the contemporary hallmarks of cancer and anti-angiogenic factors have already been approved for the treatment of several cancer types. This review aims to summarize, in a histotype-specific manner, the most current available data on the angiogenic factors implicated in SGC angiogenesis, in order to highlight the differences between the most common SGC histotypes and the factors that may have a potential role as therapeutic targets.

## 1. Introduction

Salivary gland tumors (SGTs) constitute a group of uncommon and histologically diverse neoplasms that represent 3–6% of head and neck tumors, with malignancy corresponding to 10–17% of them [1]. Most of the major-SGTs (80% of which arise from the parotid gland) are benign, whereas about half of the minor-SGTs (most of which occur in the palate) are malignant [2,3]. A slight overall female predominance of SGTs has been reported with a male to female ratio rising to 1:1.8 in some countries [4]. SGTs occur mainly in adulthood, while patients with malignancies are usually older than those with benign tumors [5].

The recent WHO classification system of 2017 recognizes a total of thirty one—11 benign and 20 malignant—distinct histotypes of SGTs [6]. In a large Japanese cohort of 5.015 SGTs, Pleomorphic Adenoma (PA, 68%) was the most frequent histotype in the group of benign SGTs, followed by Warthin tumor (WT, 26%) and Basal cell adenoma (BCA, 3%), while in the group of salivary gland carcinomas (SGCs), Adenoid Cystic (ACC, 27%) and Mucoepidermoid carcinoma (MEC, 26%) were the two most frequent histotypes, followed by Carcinoma ex PA (CXPA, 11%), Adenocarcinoma Not Otherwise Specified (AdNOS, 7%), and Acinic Cell carcinoma (AcCC, 6%) [3].

SGCs show a global annual incidence of 0.5 to 2 per 100,000 people [2] with rather high 5-year overall (OS) and disease-free (DFS) survival probabilities reported, that however seem to drop significantly in the long term [7]. Complete surgical resection with adequate free margins remains the primary treatment option for SGCs [8]. However, locally aggressive, recurrent and/or metastatic SGCs—notoriously resistant to systemic therapy [9]—are not uncommon; distant metastases are diagnosed in 25–55% of SGC patients with ACC being the most frequent (60%) histotype observed in metastatic disease [10].

De novo SGC formation as also the malignant transformation of benign SGTs is a complex phenomenon and depends on a variety of different elements, most crucial of which are reported to be cell cycle regulators such as cyclins, tumour suppressors and transcription factors, histotype-specific oncogenes such as mucoepidermoid carcinoma translocated-1 (MECT1) for MEC and v-myb avian myeloblastosis viral oncogene homolog (MYB) for ACC, various proteins including β-catenin, defensins, tenascin, and mucins, the adipocytokines leptin, ghrelin and adiponectin and finally angiogenesis factors, such as Vascular Endothelial Growth Factor (VEGF) and CD105 [11].

Angiogenesis is characterized as the biological process of blood vessel growth, regulated by the fine balance between the action of angiogenic and anti-angiogenic mediators that are produced by different cell populations such as endothelial cells (ECs), pericytes, fibroblasts, and macrophages [12]. Simplified, the process evolves into five steps starting with (1) an angiogenic stimuli such as hypoxia that increases EC permeability and cellular proliferation, followed by (2) activation of matrix metalloproteinases (MMPs), basement membrane proteolysis and degradation of the extracellular matrix (ECM), (3) activation, proliferation and migration of the ECs, (4) lumen and capillary channels formation, and finally (5) stabilization of the newly formed vessels [13].

Angiogenesis is considered as one of the contemporary hallmarks of cancer. Tumors cannot grow beyond 2–3 mm^3^ nor metastasize without new vasculature [14]. Due to rapid growth they show increased needs for oxygen, nutrients’ supply and metabolic waste drainage served by neovascularization, mainly through sprouting angiogenesis (SA), as well as through other alternative or complementary to SA mechanisms, such as vasculogenesis (detected in post-radiation tumour growth) and vascular mimicry (vessel-like structures de novo formed by malignant cells, without endothelial lining) [15]. Apart from tumor progression, angiogenesis induces invasiveness and metastasis [11]. Therefore targeting angiogenic factors is hitherto considered to be an effective way of inhibiting tumor growth [16] and restrict its metastatic potential (Figure 1).

## 2. Angiogenesis’ Evaluation

The most common method used to evaluate tumor angiogenic activity is the estimation of microvessel density (MVD) in tissue specimens, namely the number of vessels per high power field at “hot spots” within the mass. “Hot spots” are highlighted by the immunohistochemical (IHC) detection of specific ECs-markers, such as CD31 and CD34 (pan-EC markers) or even better CD105. The latest, also known as endoglin, is an activated ECs marker, part of the transforming growth factor β receptor complex that plays a key role in tumor-induced angiogenesis [17]. However, the evaluation of tumor angiogenic activity by MVD assessment lies under the constriction that not all microvessels detected necessarily contribute to tumor perfusion [18].

The VEGF family consists of VEGF-A, -B, -C, D, -E, and placenta growth factor (PGF). VEGF-A is considered to be the pivotal angiogenic signaling protein (referred to as VEGF from now on) and binds to two receptors, VEGFR-1 and -2, that play the role of negative and positive angiogenesis regulator respectively [18]. VEGF can be subjected to alternative exon splicing, leading to multiple isoforms with diverse functions, e.g., VEGF165b that, when binding to VEGFR2, induces an impaired angiogenic response [19]. It should also be underlined that VEGF secreted by tumor cells and stroma leads to the proliferation and formation of endothelial cells, which, however, may show abnormal structure and leakage [20].

Assuming that the assessment of a SGC’s neovascularization may correlate with its prognosis and that inhibition of angiogenesis may represent a potent therapeutic regiment for the management of SGC patients, this review aims to summarize recent data on the expression of biomarkers with angiogenic activity in different SGC histotypes, focusing on the most frequent of them.

## 3. Angiogenic Factors in the Most Common SGC Histotypes

### 3.1. ACC

ACC is the most common type of SGC, being characterized by slow but persistent growth, multiple recurrences and high incidence of distant metastases due to its tendency towards perineural invasion and hematological spread, resulting in poor patients’ prognosis [2,18]. Studies investigating the most common factors implicated in ACC angiogenesis are summarized in Table 1.

#### 3.1.1. MVD

Several studies have assessed MVD (both intratumoral (IMVD) and peritumoral MVD (PMVD)) in ACCs mainly by IHC staining with CD105 and/or CD34 and CD31. It should be noted that normal SG (NSG) tissues, when available, were CD105 and/or CD34 negative in almost all studies [21,22,30]. Researchers report an increased frequency of CD105 positive staining, ranging from about 8% [21] to 65% [22]. MVD was higher from that of benign lesions such as PAs [21,22,23,24,25,28,31,42,43], lower when compared to other SGC histotypes like MEC and CXPA and almost equivalent with MVD of SGC histotypes encompassing myoepithelial cells such as EMEC [21,22,23,24,25,26,28,31,33,42,43]. Interestingly, numerous studies support the presence of myoepithelial cells inhibits angiogenesis [21,22,23,25,26]. CD105 positivity was reported to be almost restricted to metastasizing ACCs [21], while MVD was found increased in metastatic ACCs [35]. However, two other studies detected similar MVD levels between non-metastasizing and metastasizing cases [21,26]. Additionally, high MVD was shown to significantly correlate with the clinical TNM stage, as well as with perineural and vascular invasion [37], while it aroused as an independent prognosticator, associated with shorter OS in one study [30]. Finally, researchers reported that ACC cells tend to form large hypo-vascularized aggregates, surrounded by large CD105 positive vessels [22], while MVD was found elevated in solid rather than cribriform and tubular types [30,32], the latest showing more tortuous vessels [27].

#### 3.1.2. VEGF

Elevated VEGF IHC expression has been shown in ACCs [28,29,30,32,33,34,35,38], especially those with a solid and trabecular pattern [28,30] with a selective localization in tubular structures noted [29]. VEGF expression was significantly higher in ACCs when compared to benign SGTs that predominantly showed weak VEGF expression [28,29]. Moreover, VEGF expression was reported to significantly and proportionally correlate with greater tumor size, advanced stage, vascular invasion and disease recurrences and metastasis, arising as an independent prognostic factor [30,34], although other studies failed to find significant differences between non metastasizing and metastasizing cases [29].

#### 3.1.3. Other Factors

(i) Nuclear Factor κB p65 subunit (NF-κB p65), when activated, upregulates several targets, including VEGF and inducible nitric oxide synthase enzyme (iNOS), the latter leading to elevated tumoral vascular density. Both NF-κB p65 and iNOS have been found to significantly correlate with MVD and VEGF levels in ACC, being higher in solid than cribriform and tubular type and arising as independent prognosticators. It should be noted that in normal SGs studied, nuclear localization of NF-κB p65 was not detected, while iNOS was faint in some salivary ducts, but not in the parenchyma [30]. The same group of researchers suggested that ACCs cells with higher metastasis feature might present greater angiogenic ability and that inhibition of NF-κB signaling, not only suppresses VEGF and iNOS expression but also affects EC mobility in ACC cell lines [44,45].

(ii) Extracellular Matrix Metalloproteinase Inducer (EMMPRIN), a transmembrane glycoprotein of the immunoglobulin superfamily that stimulates the expression of MMPs and VEGF in tumoral stromal cells, was found to overexpress in ACCs [31,32], especially in solid rather than cribriform and tubular types and aroused as an independent adverse prognosticator for OS [32].

(iii) Mutated p53 gene has been shown as a potent stimulant of VEGF, presenting an angiogenic effect additionally to its anti-apoptotic action. It was detected in ~10% of a series of ACCs and correlated with higher IMVD and VEGF expression comparing to p53(-) cases [33]. In another study, p53 expression was high in 60% of the cases and significantly correlated with VEGF levels [34].

(iv) Neuropilins−1 and −2 (Nrp−1 and −2) are members of a non–tyrosine kinase transmembrane glycoprotein family that has been reported to contribute to tumor angio- and lymphangiogenesis [35]. Nrp-2 expression in ACC, being higher in the solid than in cribriform and tubular types, was found to significantly correlate with MVD, tumor size, TNM clinical-stage, vascular invasion, and metastasis. Additionally, when ACC cell lines were treated with Nrp2 antibodies, the migration of ECs and the formation of the tubular-like structures decreased [36]. High Nrp-1 levels were associated with VEGF overexpression and elevated MVD and, along with lower levels of semaphorins 3A and 3F (Sema 3A and 3F, proteins that can inhibit tumor angiogenesis when binding to Nrps), were associated with metastasizing cases [35].

(v) Ephrin receptor EPHA2 and its ligand ephrinA1 were found elevated with high staining intensity in tumors compared to normal tissues and their levels independently correlated with MVD. Additionally, EPHA2, ephrinA1 and MVD were significantly higher in the solid than the cribriform and tubular types, although no significant difference was found between the last two types. Moreover, all of them correlated significantly with the TNM stage, perineural invasion and vascular invasion, although correlation with the clinical outcome was not assessed due to the short follow-up period. Finally, using EPHA2-Fc to block the ephrinA–EPHA2 interaction disrupted the angiogenesis process in vivo [37].

(vi) A t (6;9) translocation involving MYB and nuclear factor I/B (NFIB) genes results in the MYB-NFIB chimeric gene with transcription-regulating functions that have been intensively studied in ACC cases. Ono and Okada focused on its implication in tumor angiogenesis and proliferation. Researchers detected the chimeric gene in 34.6% of the cases significantly correlated its expression with elevated MVD and VEGF expression and suggested a possible association with the age of tumor onset [38].

(vii) The mammalian target of rapamycin (mTOR), a serine/threonine kinase multi-way implicated in carcinogenesis, has been shown to mediate tumor angiogenesis as well. Yu et al. showed that mTOR activation [as indicated by phosphorylated substrate-S6 (p-S6) overexpression] promotes angiogenesis in ACC through the Epidermal growth factor receptor (EGFR)/Signal transducer and activator of transcription-3 protein (p-Stat3) and Hypoxia-inducible factor-1α (HIF-1α)/Plasminogen activator inhibitor (PAI) pathways (activation of EGFR results in activation of Stat-3 and HIF-1a induces PAI production). All factors mentioned were overexpressed in ACCs—being only weakly or mildly expressed in NSGs and PAs—and significantly correlated with MVD, as well as with tumor cell proliferation rates. Additionally, mTOR inhibition with rapamycin effectively suppressed tumor growth by down-regulating both pathways in vitro and in vivo [39].

(viii) EGFR (a transmembrane receptor) HIF-1a (the main transcription factor mediating angiogenesis), CD31 and CD146 (a structural component of interendothelial junctions, associated with pathologic tumoral angiogenesis possibly by inducing endothelial permeability) was found to be overexpressed and significantly correlate with each other in ACC when compared with PA and NSG. Therefore, the researchers suggested that EGFR may play a role in tumor-induced angiogenesis. Let it be noted that EGFR staining was stronger in cribriform and tubular rather than in solid types, HIF-1a positivity was almost restricted in the nucleus, while CD146 staining was higher at the membrane of cells infiltrating the stroma and in the inner epithelial cells of tumoral ducts and nests of the tubular and cribriform types, respectively [40].

(ix) Epiregulin, an EGFR ligand, was found overexpressed in ACC cases compared to NSG, as well in the ACC cell line showing high lung metastasis incidence compared to the parental ACC cell line. Researchers also showed that high levels of epiregulin promoted the production of angiogenic factors such as VEGF both in ACC cell lines and in ACC tissue specimens, as well as in human pulmonary microvascular endothelial cells (HPMECs), concluding that epiregulin enhances angiogenesis in the primary tumor microenvironment and vascular permeability in the pre-metastatic lung microenvironment. Finally, they established significant correlations between high epiregulin expression and primary tumor size and stage, local recurrence, lung metastasis incidence, OS, and metastasis-free survival (MFS) [41].

### 3.2. MEC

MEC—the second most common type of SGC—is a locally invasive tumor with a solid, cystic, or microcystic pattern, consisting of variable proportions of epidermoid, intermediate and mucin-secreting cells. MECs can be of low, intermediate, or high grade, with their clinical aggressiveness proportionally increasing with the grade of malignancy [18,46]. Studies investigating the most common factors implicated in MEC angiogenesis are summarized in Table 2.

#### 3.2.1. MVD

CD105 staining and MVD is reported to be high for MECs [21,22,26,42,47,48], higher from all the other SGC types they have been compared to. For example, Cardoso et al. reported a staining frequency of 85.0%, with all the metastasizing cases studied were positive for CD105 [21], while Tadbir et al. reported CD105 positivity in 83% with tumor cells forming small aggregates surrounded by small CD105 positive vessels [22]. Costa et al. detected a significantly higher CD34-MVD and CD105-MVD in intratumoral regions compared to ACCs and EMECs with the positive vessels, usually forming a rim of capillaries immediately adjacent to the carcinomatous aggregates [26]. Advanced clinical stage, higher grade, and minor SGTs’ origination were correlated with higher MVD, while increased MVD was associated with recurrences, as indicated by a shorter DFS [48]. Contrariwise, Gleber-Netto et al. associated low IMVD with recurrence and lymph node metastasis, rendering the finding to impaired angiogenesis [47] and Luukkaa et al. reported that higher MVD predicted better prognosis [27].

#### 3.2.2. VEGF

Various researchers have reported an increased VEGF-positive immunoreaction in less differentiate versus low-grade MEC cases [28,34,49]. VEGF expression was found mainly present in epidermoid and intermediate cells, being mild or absent in mucous cells and higher in tumors of advanced stage [28]. Moreover, in two studies, VEGF expression was significantly associated with tumor differentiation, size, and relapse [34,49]. In one of them, VEGF levels correlated with lymph node metastasis, perineural and vascular invasion and clinical stage as well, arising as an independent adverse prognosticator [34], while the other failed to show any correlation with lymph node or distant metastasis [49]. Other researchers associated low VEGF expression (as well as low MVD) with tumor recurrence and nodal metastasis, implying that impaired angiogenesis could lead to an aggressive phenotype [47].

#### 3.2.3. Other factors

(i) Caveolin-1 is a protein highly expressed in endothelial cells, reported to be down-regulated via the proliferative phase and also up-regulated during the differentiation phase of angiogenesis [50]. Shi et al. found an adverse association between caveolin-1 levels and MVD, the latest arising as an adverse prognosticator correlated with stage III and IV tumors and therefore suggested that caveolin-1 may function as a tumor suppressor [48].

### 3.3. Other Histotypes

#### 3.3.1. CXPA

CXPA is an aggressive, often metastasizing SGC that develops from primary or recurrent PA. Soares et al. revealed an angiogenic switch during the progression from PA to CXPA, as indicated by gradually increasing CD105- (but not CD34-) MVD. They also concluded that CXPA with myoepithelial differentiation showed a significantly lower number of CD105 positive vessels [51], a finding that is consistent with similar studies of other SGC histotypes [21,23,26,28]. High MVD values accompanied by a moderate VEGF expression that was higher in larger tumors were also reported [28].

#### 3.3.2. AdNOS

AdNOS represents a spectrum of epithelial SGCs forming ductal and/or glandular structures other than any of the known epithelial SGCs. High CD34-MVD has been reported for AdNOS that was shown to exhibit the richest vascularization amongst other histotypes it was compared to. Additionally, intense IHC VEGF expression has been noted, especially in high-grade cases [28].

#### 3.3.3. AcCC

AcCC is a low-grade malignancy usually occurring in the parotid gland that shows a tendency to recur and metastasize. Margaritescu et al. showed active angiogenesis in AcCCs, supported by elevated CD105-MVD scores and the IHC reactivity of tumor cells and ECs for VEGF and its receptors (VEGFR1 and VEGFR2). The highest CD105-MVD score was recorded in the stable variant, while VEGF’s highest reactivity was mainly recorded in intercalated duct-like and nonspecific glandular cells and the microcystic and stable variants [52]. Elevated CD34-MVD has also been reported [28].

#### 3.3.4. EMEC

EMEC is a usually low-grade SGC that combines epithelial and myoepithelial components. Costa et al. reported low CD34- and CD105-MVD in EMECs that are compatible with the hypothesis that the presence of myoepithelial cells inhibits angiogenesis [26].

#### 3.3.5. Polymorphous Adenocarcinoma (PAC, ex PLGA)

PAC derives mainly from the minor SGs and is characterized by cytological uniformity and architectural diversity, showing a generally good prognosis, although high-grade transformation has been reported. Cardoso et al. reported a frequency of CD105 positive staining in 42.1% of PLGAs studied with positivity restricted to non-metastasizing cases [21].

## 4. Anti-Angiogenic Factors Studied for SGCs Therapy

Anti-angiogenic factors have already been approved for the treatment of several cancers, such as thyroid, breast, non-squamous non-small cell lung, gastric, colorectal, hepatocellular, pancreatic, renal cell, cervical and ovarian epithelial cancer [53,54]. As mentioned above, locally aggressive, recurrent, and/or metastatic SGCs are not uncommon and frequently show resistance to systemic therapy. The effect of various chemical substances with anti-angiogenic activity has been evaluated in SGCs, with research being focused on ACC, since it is one of the two most frequent SGC histotypes and the majority of ACC patients die because of local recurrences and/or distant metastases [2,18], while the contribution of surgery and radiation in patients’ management is limited in the long term [55].

### 4.1. Preclinical Trials

(a) Isoliquiritigenin (ISL) is a licorice derived flavonoid studied for its inhibiting effect on ACC induced angiogenesis in vitro, ex vivo, and in vivo [56]. ISL was found to inhibit tumor-induced but not normal preexisting angiogenesis in a concentration-dependent manner, an effect that was more intense in high compared to low metastasis ACC cell lines. It also had a restrictive effect on VEGF levels by downregulating the mammalian target of the rapamycin (mTOR) pathway, which led to a significant decrease of MVD within xenograft tumors.

(b) AEE788 is a dual EGFR and VEGFR Tyrosine Kinase Inhibitor (TKI) studied both in vitro and in vivo for its anti-tumor effect, alone or in combination with chemotherapy (cisplatin or paclitaxel) [57]. It was found to reduce EGFR and VEGFR-2 phosphorylation, MVD and MMP-9 and MMP-2 expression and enhance tumor cell and EC apoptosis. AEE788 was also shown to restrict the incidence of vascular metastasis in orthotopic nude mouse models of human ACC, while all its effects were amplified when combined with chemotherapy.

(c) Vandetanib is another dual EGFR and VEGFR TKI evaluated for its anti-tumor effect both in vitro and in vivo [58]. Researchers noted a dose-dependent restriction of VEGFR-2 and EGFR phosphorylation, as well as of cell proliferation and an enhancement of cell apoptosis in ACC cell lines. Additionally, a tumor volume and MVD reduction, accompanied by increased tumor cell and EC apoptosis were found in tumor xenografts.

### 4.2. Clinical Trials

(a) AG-013736 (axitinib) is an orally administered VEGFR, platelet-derived growth factor (PDGF)-b and v-Kit TKI. Its activity in advanced solid tumors—amongst them, one ACC—was studied in a Phase I trial [59]. The ACC patient showed a partial response after three cycles that lasted for four months; however, treatment was discontinued due to intolerable side effects. Axitinib was also evaluated in a Phase II study of 33 patients with incurable ACC of any primary origin (78.7% of the cases derived from the SGs) [60]. Although the study failed to achieve its primary endpoint (best OS), Axitinib induced tumor shrinkage in most of the patients. Moreover, genomic analysis implied that ACCs with 4q12 amplification might present a subgroup that can benefit from TKI based therapy. Finally, Axitinib’s activity was studied in a Phase II study of 26 patients with recurrent and/or metastatic SGC of the upper aerodigestive tract, including 6 ACC cases and 20 non-ACC cases (5 AdNOS, 5 Poorly differentiated carcinomas, 3 AcCCs, 2 Clear cell carcinomas, 1 PAC, 1 SDC, 1 EMEC, 1 CXPA, and 1 MEpC) [61]. The response rate was 8% with only two partial responses (1 patient with ACC and 1 with poorly differentiated carcinoma), and thus this trial failed to meet its primary endpoint of >3 responses.

(b) Sunitinib is a multi-targeted TKI (VEGFR included) evaluated in a Phase II study of 14 patients with progressive, recurrent and/or metastatic ACC, one of which discontinued therapy during the first cycle due to toxicity [62]. Although there were no objective responses noted, the drug was rather well-tolerated, while a prolonged period of stable disease (≥6 months) was noted in 62% of the patients with a median time to progression of 7.2 months.

(c) Sorafenib is another a multi-targeted TKI (VEGFR included). Its activity was evaluated in a Phase II study of 23 patients with unresectable locally recurrent and/or metastatic ACC [63]. Sorafenib administration was followed by side effects resulting in dose reduction in 17 out of the 23 patients and median progression-free (PFS) and overall (OS) survival rates were rather modest (11.3 and 19.6 months respectively). The drug was also studied in a Phase II trial that enrolled 37 patients with recurrent and/or metastatic SGC, including 19 ACCs, 5 high-grade MECs, 7 AdNOS, 2 SDCs, 3 MEpCs, and 1 Poorly differentiated carcinoma [64]. Researchers noted an incidence of severe side effects of 29.7%. Median PFS and OS for ACC patients were 8.9 and 26.4 months, respectively, versus 4.2 and 12.3 months, respectively for the group of non-ACC cases. Out of the six objective responders, 2 suffered ACC, 1 MEC, 1 AdNOS, 1 SDC, and 1 poorly differentiated carcinoma. Additionally, regarding ACCs, MYB protein expression was noted in 94% and the MYB-NFIB fusion oncogene was detected in 64%.

(d) Pazopanib is an oral inhibitor of VEGFR, PDGFR, and KIT. Its efficiency is being evaluated in a Phase II trial studying 63 patients with progressive, recurrent, or metastatic SGC (45 ACCs and 18 non-ACCs) [65]. Researchers report a significant decrease in tumor growth rates, foreseeing a promising efficiency of the drug.

(e) Lenvatinib is an oral, TKI approved for the treatment of radioiodine-refractory thyroid cancer and unresectable hepatocellular carcinoma with important inhibitory activity against VEGFR1, VEGFR2, VEGFR3, fibroblast growth factor receptors (FGFRs) 1–3, KIT, PDGFRα and β. Its activity was evaluated in a Phase II study of 32 patients with ACC. Median PFS time was 17.5 months (95% CI, 7.2 months to not reached), although only eight progression events were observed. These are the best results among all the VEGFR-targeting TKI studies regard to ACC [66].

(f) Dasatinib is an oral aminothiazole analog and has inhibition specificity for five kinases/kinase families (BCRABL, c-SRc, c-KIT, PDGFβ receptor, and EPHA2). Its efficiency is being evaluated in a Phase II trial study with 54 patients (40 ACC, 14 non-ACC). Median PFS was 4.8 months. Median OS was 14.5 months. For 14 assessable non-ACC patients, none had an objective response, triggering the early stopping rule [67].

(g) Regorafenib is a TKI that targets VEGFR, FGFR, and PDGFR evaluated in a Phase II study of 38 patients with progressive, recurrent and/or metastatic ACC. There are two RECIST v1.1 evaluable primary endpoints: (1) the proportion of patients alive at six months without progression of the disease, (2) best overall response rate (ORR). Unfortunately, the study failed to meet these endpoints, but regorafenib may elicit disease control for a subset of ACC patients [68].

(h) Dovitinib inhibits the VEGFR, PDGFR, c-Kit, CSF-1R, RET, TrkA, FLT3 receptor kinases, and FGFRs 1–3. Its activity in metastatic and/or unresectable ACC tested in two Phase II studies with 32 and 35 patients, respectively. At the first study, the 4-month PFS probability was 80.4%, and the median PFS was 6.0 months, also was observed shrink of the tumor in 22 patients (68.8%). Finally, a partial response had 1 patient [69]. At the second trial, the primary endpoints were ORR and tumor growth rate. PFS, OS metabolic response, biomarker, and quality of life were secondary endpoints. About the results, the median PFS was 8.2 months and OS was 20.6 months. Very significant was the reduction of tumor growth rate (1.95 to 0.63) [70]. Table 3 represents clinical trials with more anti-angiogenic agents.

## 5. Conclusions

SGCs constitute a group of rare and histologically diverse neoplasms with precise diagnosis and patients’ management frequently being quite challenging. Contemporary research has upgraded angiogenesis as one of the hallmarks of cancer [14] and has led to the development of pharmacological anti-angiogenetic agents, primarily through the blockage of VEGF/VEGFR signaling [54]. Some of them have already been approved for the treatment of several cancer types.

Current data shows that angiogenesis may play an essential role in the progression of SGCs. In the studies reviewed, angiogenetic factors like VEGF, were overexpressed in SGCs and frequently significantly associated with clinical stage, histological grade, recurrence, and survival/prognosis and thus, their inhibition could represent a potential therapeutic target and should be thoroughly investigated. Additionally, different SGC histotypes seem to show different patterns of angiogenesis. Therefore, there is an urgent need for carefully designed studies in large and histotype-specific groups in order to broaden the spectrum of therapeutic regimens, especially for advanced SGCs.

## Figures and Tables

**Figure 1 ijms-21-09335-f001:**
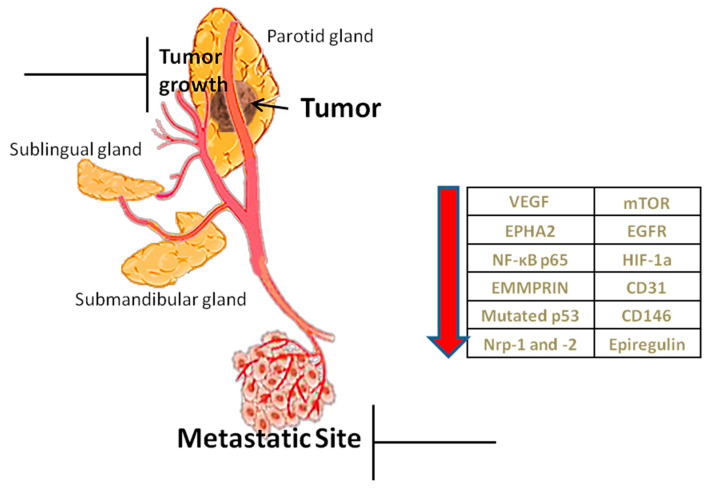
Role of angiogenesis and factors implicated. Angiogenesis helps in the growth of the tumor through the transfer of nutrients but also in its metastasis. The table shows angiogenic factors, which, when reduced by inhibitors, prevent tumor growth and metastasis.

**Table 1 ijms-21-09335-t001:** Studies investigating the most common factors implicated in Adenoid Cystic (ACC) angiogenesis.

N° of ACC Cases	Other SGC Histotypes Included * (N^o^ of Cases)	EC Markers and Other Factors ** (ICH Detected, If Not Otherwise Specified)	Vs NSG (N° of Cases)	Vs Benign SGTs (N° of Cases)	Significant Correlations with Clinicopathological Parameters	Ref.
51	MEC (40), PLGA (19)	CD105, IMVD	Yes (83, near SGTs)	PA (29)	CD105 (+) vessels restricted to metastasizing cases	[21]
19	MEC (20)	CD105, IMVD, Ki67	Yes (10)	PA (20)	-	[22]
9	MEC (8), MEpC (1)	CD31, CD105, MVD	-	PA (21), WT (2), BCA (2)	Only CD105-MVD differed between benign and malignant SGTs	[23]
5	MEC (6), SDC (4)	CD34, MVD	-	PA (15)	-	[24]
20	MEC (20)	CD105, MVD	Yes (10)	PA (20)	-	[25]
31	MEC (37), EMEC (14)	CD34, CD105, IMVD, PMVD, Vimentin, α-SMA, Ki67, Prdx-1	-	-	-	[26]
37	MEC (18)	CD34, MVD	-	-	-	[27]
4	MEC (6), CXPA (6), AdNOS (4), AcCC (5)	VEGF, CD34, MVD,	Yes (near SGTs)	PA (8), WT (7), BCA (5)	-	[28]
50	MEC (40), PLGA (19)	VEGF, TP	-	PA (30)	-	[29]
80	-	CD34, MVD, VEGF, NF-κB p65, iNOS	Yes (20)	-	CD34-MVD, VEGF, NF-κB p65 and iNOS: independent prognosticators for OS	[30]
33	MEC (25)	CD34, MVD, EMMPRIN (ICH & RT-PCR in frozen sections)	Yes (9)	PA (28)	-	[31]
72	-	CD34, MVD, EMMPRIN, VEGF, Ki67, MMP −2 and −9	Yes (20)	-	EMMPRIN (+): independent prognosticator for OS	[32]
11	MEC (10), AcCC (7), SCC (3)	CD34, IMVD, VEGF, p53	-	-	-	[33]
15	MEC (14), AdNOS (6), PLGA (4), CXPA (5), SDC (2)	VEGF, p53, Ki67	-	-	VEGF expression with p53 expression, tumor size, lymph node metastasis, perineural and vascular invasion, clinical stage and recurrence VEGF expression: independent prognosticator for OS	[34]
60 ^#^	-	NRP1 and 2, VEGF, Sema-3A, Sema-3F, CD31, D240	Yes (30)	-	NRP1, VEGF and MVD: greater in metastatic ACC	[35]
50	-	NRP2, CD34, MVD	Yes (20, near SGTs)	-	NRP2 expression with TMN, clinical stage, vascular invasion and metastasis	[36]
49	-	CD34, MVD, S100 and p-Tyr (IHC), EPHA2 and ephrinA1 (ICH, RT-PCR and Western blotting)	Yes (10)	-	EPHA2/ephrinA1 levels and MVD with clinical TNM stage, perineural and vascular invasion	[37]
26	-	*MYB-NFIB* chimeric gene (RT-PCR and direct sequencing), CD31, VEGF and Ki67 (IHC)	-	-	-	[38]
72	-	CD34, MVD, Ki67, p-S6^S235/236^, EGFR, p-Stat3^T705^, HIF-1α and PAI^●^	Yes (18)	PA (12)	-	[39]
74	-	CD31, MVD, EGFR, CD146, HIF-1a	Yes (18)	PA (12)	-	[40]
167 ^##^	-	Epiregulin, CD31, CD34	Yes (52 ^§^)	-	Epiregulin levels with tumour size and stage, local recurrence, lung metastasis, OS and MFS	[41]

* Abbreviations for other SGC histotypes included. MEC: Mucoepidermoid Carcinoma, PLGA: Polymorphus Low Grade Adenocarcinoma, MEpC: Myoepithelial Carcinoma, SDC: Salivary Duct Carcinoma, EMEC: Epithelial-myoepithelial Carcinoma, CXPA: Carcinoma ex Pleomorphic Adenoma, AdNOS: Adenocarcinoma Not Otherwise Specified, AcCC: Acinic Cell Carcinoma, SCC: Squamous Cell Carcinoma. ** Abbreviations for EC markers & other Factors. MVD: Microvessel Density, IMVD: Intratumoral MVD, PMVD: Peritumoral MVD, α-SMA: α-Smooth Muscle Actin, Prdx-1: peroxiredoxin-I, VEGF: Vascular Endothelial Growth Factor, TP: Thymidine phosphorylase, NF-κB p65: Nuclear Factor κB p65 subunit, iNOS: inducible Nitric Oxide Synthase, EMMPRIN: Extracellular Matrix Metalloproteinase Inducer, RT-PCR: Reverse Transcription Polymerase Chain Reaction, MMP: metalloproteinase, NRP: Neuropilin, Sema: Semaphorin, p-Tyr: Phosphotyrosine, EPHA2: Ephrin receptor A2, MYB-NFIB chimeric gene: v-myb avian myelobastosis viral oncogene homolog-nuclear factor I/B chimeric gene, p-S6: phosphorylated substrate-S6, EGFR: Epidermal Growth Factor Receptor, p-Stat3: Signal transducer and activator of transcription-3 protein, HIF-1α: Hypoxia-Inducible Factor-1α, PAI: Plasminogen Activator Inhibitor. ^#^ 30 metastasizing and 30 non metastasizing. ^##^ 107 paraffin-embedded and 60 frozen tissues. p-S6^S235/236^, EGFR, p-Stat3^T705^, HIF-1α and PAI were also studied by Western blotting and immunofluorescence in ACC cell lines and by Western blotting and immunohistochemical (IHC) in nude mice xenografts. ^§^ 11 paraffin-embedded and 11 frozen tissues. Researchers also examined the association between epiregulin’ s expression and lung metastasis of ACC in ACC cell lines and mice xenografts, as well as the biological effects of epiregulin-enriched exosomes.

**Table 2 ijms-21-09335-t002:** Studies investigating the most common factors implicated in Mucoepidermoid carcinoma (MEC) angiogenesis.

N° of MEC Cases	Other SGC Histotypes Included * (N° of Cases)	EC Markers and Other Factors ** (ICH Detected, If Not Otherwise Specified)	Vs Normal SG (N° of Cases)	Vs Benign SGTs (N° of Cases)	Significant Correlations with Clinicopathological Parameters	Ref.
40	ACC (51), PLGA (19)	CD105, IMVD	Yes (83, near SGTs)	PA (29)	-	[21]
20	ACC (19)	CD105, IMVD, Ki67	Yes (10)	PA (20)	-	[22]
8	ACC (9), MEpC (1)	CD31, CD105, MVD	-	PA (21), WT (2), BCA (2)	-	[23]
6	ACC (5), SDC (4)	CD34, MVD	-	PA (15)	-	[24]
20	ACC (20)	CD105, MVD	Yes (10)	PA (20)	-	[25]
37	ACC (31), EMEC (14)	CD34, CD105, IMVD, PMVD, Vimentin, α-SMA, Ki67, Prdx-1	-	-	-	[26]
18	ACC (37)	CD34, MVD	-	-	-	[27]
6	ACC (4), CXPA (6), AdNOS (4), AcCC (5)	CD34, MVD, VEGF	Yes (near SGTs)	PA (8), WT (7), BCA (5)	-	[28]
40	ACC (50), PLGA (19)	VEGF, TP	-	PA (30)	-	[29]
26	-	CD105, D2-40, MVD, LVD, VEGF-A and VEGF-C	-	-	High IMVD in younger patients Low VEGF-A and MVD with recurrence and nodal metastasis	[47]
70	-	CD34, MVD, VEGF, iNOS	Yes (40)	-	iNOS and VEGF expression with tumor differentiation, size, metastasis and relapse	[49]
25	ACC (33)	CD34, MVD, EMMPRIN (ICH and RT-PCR in frozen sections)	Yes (9)	PA (28)	-	[31]
10	ACC (11), AcCC (7), SCC (3)	CD34, IMVD, VEGF, p53	-	-	-	[33]
14	ACC (15), AdNOS (6), PLGA (4), CXPA (5), SDC (2)	VEGF, p53, Ki67	-	-	VEGF expression with p53 expression and higher grade, tumor size, lymph node metastasis, perineural and vascular invasion, clinical stage and recurrence VEGF expression: independent prognosticator for OS	[34]
75	-	Caveolin-1, CD34, IMVD, VEGF	-	-	Decreased caveolin−1 expression rates with tumors of shorter duration, stage III and IV tumours and recurrent disease MVD higher in stage III and IV tumours and independent adverse prognosticator	[48]

* Abbreviations for other SGC histotypes included. ACC: Adenoid Cystic Carcinoma, PLGA: Polymorphus Low Grade Adenocarcinoma, MEpC: Myoepithelial Carcinoma, SDC: Salivary Duct Carcinoma, EMEC: Epithelial-myoepithelial Carcinoma, CXPA: Carcinoma ex Pleomorphic Adenoma, AdNOS: Adenocarcinoma Not Otherwise Specified, AcCC: Acinic Cell Carcinoma, SCC: Squamous Cell Carcinoma. ** Abbreviations for EC markers & other Factors. MVD: Microvessel Density, IMVD: Intratumoral MVD, PMVD: Peritumoral MVD, α-SMA: α-Smooth Muscle Actin, Prdx-1: peroxiredoxin-I, VEGF: Vascular Endothelial Growth Factor, TP: Thymidine phosphorylase, LVD: Lymphatic Vessel Density, iNOS: inducible Nitric Oxide Synthase, EMMPRIN: Extracellular Matrix Metalloproteinase Inducer.

**Table 3 ijms-21-09335-t003:** Current clinical trials relevant to angiogenesis and salivary gland malignant tumors.

NCT Number	Title	Status	Study Results	Drugs	Phase
**NCT02558387**	Trial of BIBF1120 (Nintedanib) in Patients With Recurrent or Metastatic Salivary Gland Cancer of the Head and Neck	Unknown status	No Results Available	BIBF1120	II
**NCT01254617**	Lenalidomide and Cetuximab in Treating Patients With Advanced Colorectal Cancer or Head and Neck Cancer	Completed	No Results Available	Lenalidomide Cetuximab	I
**NCT00588770**	Chemotherapy With or Without Bevacizumab in Treating Patients With Recurrent or Metastatic Head and Neck Squamous Cell Carcinoma	Active, not recruiting	Has Results	Bevacizumab Carboplatin Cisplatin Docetaxel Fluorouracil	III
**NCT00492089**	Bevacizumab in Reducing CNS Side Effects in Patients Who Have Undergone Radiation Therapy to the Brain for Primary Brain Tumor, Meningioma, or Head and Neck Cancer	Completed	Has Results	Bevacizumab	II
**NCT00101348**	Erlotinib and Cetuximab With or Without Bevacizumab in Treating Patients With Metastatic or Unresectable Kidney, Colorectal, Head and Neck, Pancreatic, or Non-Small Cell Lung Cancer	Completed	No Results Available	Erlotinib hydrochloride Cetuximab Bevacizumab	I, II
**NCT00023959**	Bevacizumab, Fluorouracil, and Hydroxyurea Plus Radiation Therapy in Treating Patients With Advanced Head and Neck Cancer	Completed	No Results Available	Bevacizumab Hydroxyurea Fluorouracil	I
**NCT00005647**	SU5416 and Paclitaxel in Treating Patients With Recurrent, Locally Advanced or Metastatic Cancer of the Head and Neck	Completed	No Results Available	Paclitaxel Semaxanib	I

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
