# Peer review of "The Impact of Angiogenesis in the Most Common Salivary Gland Malignant Tumors"

_ijms, 2020, doi:10.3390/ijms21249335_

Round 1
Reviewer 1 Report
It was my great pleasure to review the manuscript “The impact of angiogenesis in the most common salivary gland malignant tumors” by Pouloudi and colleagues. The review summarizes the data on angiogenic pathways and inhibitors across various histological subtypes. The manuscript is overall well written but is missing lot of clinical information.
1. For example, the authors did not provide any information on lenvatinib (Phase II Study of Lenvatinib in Patients With Progressive, Recurrent or Metastatic Adenoid Cystic Carcinoma. PMID: 30939095) which has the best reported data among angiogenesis inhibitors in adenoid cystic carcinoma.
2. Another agent dasatinib which has antiangiogenic effects and evaluated in adenoid cystic carcinoma has not been mentioned. (Phase II trial of dasatinib for recurrent or metastatic c-KIT expressing adenoid cystic carcinoma and for nonadenoid cystic malignant salivary tumors. Ann Oncol. 2016;27:318–323 PMID: 26598548)
3. Similarly regorafenib has also been studied (Phase II study of regorafenib in progressive, recurrent/metastatic adenoid cystic carcinoma. J Clin Oncol. 2016;34(suppl 15; abstr 6096)
4. Dovitinib though not approved has also two studies
Dillon PM, Petroni GR, Horton BJ, et al. A phase II study of dovitinib in patients with recurrent or metastatic adenoid cystic carcinoma. Clin Cancer Res. 2017;23:4138–4145.
Keam B, Kim S-B, Shin SH, et al. Phase 2 study of dovitinib in patients with metastatic or unresectable adenoid cystic carcinoma. Cancer. 2015;121:2612–2617.
5. Will suggest providing a figure encapsulating the role of angiogenesis pathway in salivary gland carcinoma progression.
6. Please provide future directions and guidance for this filed. It can be done by providing a table from clinicaltrials.gov summarizing relevant studies.
Author Response
Reviewer 1
It was my great pleasure to review the manuscript “The impact of angiogenesis in the most common salivary gland malignant tumors” by Pouloudi and colleagues. The review summarizes the data on angiogenic pathways and inhibitors across various histological subtypes. The manuscript is overall well written but is missing lot of clinical information.
Authors response: Initially, we would like to thank you for the constructive comments that aim to the improvement of our manuscript. We would also like to thank you for your kind words that recognize our effort to demonstrate the critical role of angiogenesis in salivary gland malignant tumors.
- For example, the authors did not provide any information on lenvatinib (Phase II Study of Lenvatinib in Patients With Progressive, Recurrent or Metastatic Adenoid Cystic Carcinoma. PMID: 30939095) which has the best reported data among angiogenesis inhibitors in adenoid cystic carcinoma.
- Another agent dasatinib which has antiangiogenic effects and evaluated in adenoid cystic carcinoma has not been mentioned. (Phase II trial of dasatinib for recurrent or metastatic c-KIT expressing adenoid cystic carcinoma and for nonadenoid cystic malignant salivary tumors. Ann Oncol. 2016;27:318–323 PMID: 26598548)
- Similarly regorafenib has also been studied (Phase II study of regorafenib in progressive, recurrent/metastatic adenoid cystic carcinoma. J Clin Oncol. 2016;34(suppl 15; abstr 6096)
- Dovitinib though not approved has also two studies
Dillon PM, Petroni GR, Horton BJ, et al. A phase II study of dovitinib in patients with recurrent or metastatic adenoid cystic carcinoma. Clin Cancer Res. 2017;23:4138–4145.
Keam B, Kim S-B, Shin SH, et al. Phase 2 study of dovitinib in patients with metastatic or unresectable adenoid cystic carcinoma. Cancer. 2015;121:2612–2617.
Authors response: We thank the reviewer for the comments and the interesting agents and relative papers. We have made the appropriate editing and added the references.
“(e) Lenvatinib is an oral, TKI approved for the treatment of radioiodine-refractory thyroid cancer and unresectable hepatocellular carcinoma with important inhibitory activity against VEGFR1, VEGFR2, VEGFR3, fibroblast growth factor receptors (FGFRs) 1-3, KIT, PDGFRα and β. Its activity was evaluated in a Phase II study of 32 patients with ACC. Median PFS time was 17.5 months (95% CI, 7.2 months to not reached), although only eight progression events were observed. These are the best results among all the VEGFR-targeting TKI studies regard to ACC [66].
(f) Dasatinib is an oral aminothiazole analog, and has inhibition specificity for five kinases/kinase families (BCRABL, c-SRc, c-KIT, PDGFβ receptor, and EPHA2). Its efficiency is being evaluated in a Phase II trial study with 54 patients (40 ACC, 14 non-ACC). Median PFS was 4.8 months. Median OS was 14.5 months. For 14 assessable non-ACC patients, none had an objective response, triggering the early stopping rule [67].
(g) Regorafenib is a TKI that targets VEGFR, FGFR, and PDGFR evaluated in a Phase II study of 38 patients with progressive, recurrent and/or metastatic ACC. There are two RECIST v1.1 evaluable primary endpoints: 1) the proportion of patients alive at six months without progression of the disease, 2) best overall response rate (ORR). Unfortunately, the study failed to meet these endpoints, but regorafenib may elicit disease control for a subset of ACC patients [68].
(h) Dovitinib inhibits the VEGFR, PDGFR, c-Kit, CSF-1R, RET, TrkA, FLT3 receptor kinases and FGFRs 1–3. Its activity in metastatic and/or unresectable ACC tested in two Phase II studies with 32 and 35 patients, respectively. At the first study, the 4-month PFS probability was 80.4%, and the median PFS was 6.0 months, also was observed shrink of the tumor in 22 patients (68.8%). Finally a partial response had 1 patient [69]. At the second trial, the primary endpoints were ORR and tumor growth rate. PFS, OS metabolic response, biomarker, and quality of life were secondary endpoints. About the results, the median PFS was 8.2 months and OS was 20.6 months. Very significant was the reduction of tumor growth rate (1.95 to 0.63) [70].”
- Will suggest providing a figure encapsulating the role of angiogenesis pathway in salivary gland carcinoma progression
Authors response: We thank the reviewer for the suggestion and we are providing a figure with the role of angiogenesis and factors implicated.
- Please provide future directions and guidance for this filed. It can be done by providing a table from clinicaltrials.gov summarizing relevant studies.
Authors’ response: We thank the reviewer for the remark. We are providing a table with current and relevant studies.
Reviewer 2 Report
Minor revisions:
- Line 289, please add the reference.
- Please include the description of inclusion/exclusion criteria for the articles summarized in the present review.
- Please provide the details if, and to what extend the review follows the PRISMA criteria for the review articles.
Author Response
Reviewer 2
Minor revisions:
- Line 289, please add the reference.
Authors’ response: We thank the reviewer for the comment. We added the appropriate references :
- Wang Z, Dabrosin C, Yin X, Fuster MM, Arreola A, Rathmell WK, Generali D, Nagaraju GP, El-Rayes B, Ribatti D, Chen YC, Honoki K, Fujii H, Georgakilas AG, Nowsheen S, Amedei A, Niccolai E, Amin A, Ashraf SS, Helferich B, Yang X, Guha G, Bhakta D, Ciriolo MR, Aquilano K, Chen S, Halicka D, Mohammed SI, Azmi AS, Bilsland A, Keith WN, Jensen LD (2015). Broad targeting of angiogenesis for cancer prevention and therapy. Semin Cancer Biol. 35 Suppl(Suppl):S224-S243. doi: 10.1016/j.semcancer.2015.01.001
- Ramjiawan, R. R., Griffioen, A. W., & Duda, D. G. (2017). Anti-angiogenesis for cancer revisited: Is there a role for combinations with immunotherapy? Angiogenesis, 20(2), 185–204. https://doi.org/10.1007/s10456-017-9552-y
- Please include the description of inclusion/exclusion criteria for the articles summarized in the present review.
Authors’ response: We thank the reviewer for the comment. We used all the relevant papers of the English literature regarding SGC and angiogenesis.
- Please provide the details if, and to what extend the review follows the PRISMA criteria for the review articles.
Authors’ response: We thank the reviewer for the remark. This review is not a systematic or meta-analysis review; thus, it does not need a PRISMA statement.
Round 2
Reviewer 1 Report
The authors have made all the suggested changes.